# Quantification of Cancer-Developing Idiopathic Pulmonary Fibrosis Using Whole-Lung Texture Analysis of HRCT Images

**DOI:** 10.3390/cancers13225600

**Published:** 2021-11-09

**Authors:** Chia-Hao Liang, Yung-Chi Liu, Yung-Liang Wan, Chun-Ho Yun, Wen-Jui Wu, Rafael López-González, Wei-Ming Huang

**Affiliations:** 1Department of Biomedical Imaging and Radiological Sciences, National Yang Ming Chiao Tung University, Taipei City 112, Taiwan; leehomliang@gmail.com; 2Department of Radiology, School of Medicine, College of Medicine, Taipei Medical University, Taipei City 110, Taiwan; 3Department of Radiology, Wan Fang Hospital, Taipei Medical University, Taipei City 116, Taiwan; 4Department of Diagnostic Radiology, Xiamen Chang Gung Hospital, Xiamen 361028, China; gigiliu1974@gmail.com; 5Department of Imaging Technology Division, Xiamen Chang Gung Hospital, Xiamen 361028, China; 6Department of Medical Imaging and Intervention, Linkou Chang Gung Memorial, College of Medicine, Chang Gung University, Taoyuan 333, Taiwan; ylw0518@gmail.com; 7Department of Radiology, Mackay Memorial Hospital, Taipei City 104, Taiwan; med202657@gmail.com; 8Department of Medicine, Mackay Medical College, New Taipei City 252, Taiwan; 9Mackay Junior College of Medicine, Nursing, and Management, New Taipei City 252, Taiwan; 10Division of Pulmonary and Critical Care Medicine, Mackay Memorial Hospital, Taipei City 104, Taiwan; ybeei740317@gmail.com; 11Quantitative Imaging Biomarkers in Medicine, QUIBIM S.L., 46021 Valencia, Spain; lopgon.rafael@gmail.com

**Keywords:** idiopathic pulmonary fibrosis, lung cancer, radiomics, risk factors

## Abstract

**Simple Summary:**

Idiopathic pulmonary fibrosis (IPF) patients have a significantly higher risk of developing lung cancer. Traditional risk factors including age, male gender, smoking status, and emphysema have been reported. However, there are only limited data on radiomics features from HRCT images useful for risk stratification of IPF patients for lung cancer. In this study, we found that texture-based radiomics features can be differentiated between IPF patients with and without cancer development, and their diagnostic accuracy is not inferior to that of traditional risk factors. By combining radiomics features and traditional risk factors, the diagnostic accuracy can be improved.

**Abstract:**

Idiopathic pulmonary fibrosis (IPF) patients have a significantly higher risk of developing lung cancer (LC). There is only limited evidence of the use of texture-based radiomics features from high-resolution computed tomography (HRCT) images for risk stratification of IPF patients for LC. We retrospectively enrolled subjects who suffered from IPF in this study. Clinical data including age, gender, smoking status, and pulmonary function were recorded. Non-contrast chest CT for fibrotic score calculation and determination of three dimensional measures of whole-lung texture and emphysema were performed using a promising deep learning imaging platform. The results revealed that among 116 subjects with IPF (90 non-cancer and 26 lung cancer cases), the radiomics features showed significant differences between non-cancer and cancer patients. In the training cohort, the diagnostic accuracy using selected radiomics features with AUC of 0.66–0.73 (sensitivity of 80.0–85.0% and specificity of 54.2–59.7%) was not inferior to that obtained using traditional risk factors, such as gender, smoking status, and emphysema (%). In the validation cohort, the combination of radiomics features and traditional risk factors produced a diagnostic accuracy of 0.87 AUC and an accuracy of 75.0%. In this study, we found that whole-lung CT texture analysis is a promising tool for LC risk stratification of IPF patients.

## 1. Introduction

Idiopathic pulmonary fibrosis (IPF) is a chronic, progressive, fibrosing interstitial pneumonia with unknown etiology. Its prognosis is generally poor, with a median survival of 3–5 years after diagnosis [1]. The common comorbidities of IPF include lung cancer (LC), pulmonary hypertension, chronic obstructive pulmonary disease (COPD), pulmonary embolism, and pulmonary infections [1,2]. The timely identification and treatment of comorbidities play an important role in overall IPF patient survival [2].

Several studies provide epidemiologic evidence that IPF patients have a higher risk of developing lung cancer [3,4,5]. The prevalence of LC in patients with IPF was estimated to be 3.0–45.7% [3,6,7]. Kato et al. reported that the incidence of lung cancer development was 25.2 cases per 1000 person-years, and the 1-, 3-, and 5-year all-cause mortality rates after lung cancer diagnosis were 53.5%, 78.6%, and 92.9%, respectively [3]. A large retrospective study involving 870 patients showed that, among patients with IPF and surgically treated non-small-cell lung cancer, surgery-related mortality and 5-year survival rate were 7.1% and 61.6%, respectively; both were significantly poorer than those for patients without IPF (1.9% and 83.0%) [8]. These high incidence and mortality rates are crucial to understand risk stratification for LC of IPF patients.

IPF involves several LC risk factors, such as age, male gender, smoking status, and emphysema [3,5,9]. Tomassetti et al. reported that patients with LC were more frequently smokers, with combined pulmonary fibrosis and emphysema, compared with IPF-only patients (52.0% vs. 32.0%) [9]. In modern diagnostic algorithms and follow-up strategies, HRCT is important for early LC detection, and most developed lung cancers are located in the peripheral lung and lower lung lobe, as well as adjacent to tissue with usual interstitial pneumonia (UIP) [3,7,9,10]. However, limited HRCT findings are used in risk stratification for LC of IPF patients, likely due to the complexity of IPF fibrosis, the large time investment required, and the low reproducibility of manual disease segmentation of different HRCT patterns.

As imaging quantification tools improve, radiomics offers an objective quantification of tissue characteristics to help detect abnormalities in images not observable by visual evaluation only [11,12,13]. Due to its high objectivity, radiomics has a great potential to help clinical decision making and benefit diagnosis, treatment, and prognosis of multiple diseases, particularly malignant prediction [14,15,16]. In the field of interstitial lung diseases, radiomics is still developing. Martini et al. reported that radiomics findings were correlated with gender, age, and pulmonary function in patients with systemic sclerosis [11]. Stefano et al. showed that the percentage of normally attenuated lungs can help enhance IPF diagnostic processes [17]. To the best of our knowledge, no other radiomics analyses currently exist for risk stratification of IPF patients for LC. In this retrospective study, we aimed to evaluate if texture-based radiomics features can differentiate between IPF patients with and without cancer development and to compare their usefulness with that of clinical data.

## 2. Materials and Methods

### 2.1. Patient Selection

From March 2010 to the last decade of April 2021, 556 cases with reported interstitial lung diseases were enrolled in the study for further investigation. The inclusion criterion was that the diagnosis of these cases complied with the ATS/ERS/JRS/ALAT guideline of 2018 [1]. The cases before 2018 were all well reviewed using the same criteria. Since multidisciplinary discussions (MDD) started in our hospital in December 2019, for the included cases not evaluated through MDD, the IPF diagnosis was only confirmed by pulmonologists and rheumatologists when the typical UIP or probable UIP patterns on HRCT were found. The exclusion criteria were cases (1) diagnosed as non-IPF disease, including connective tissue disease related to interstitial lung disease (CTD-ILD), lymphangioleiomyomatosis (LAM), lymphocytic interstitial pneumonia (LIP), chronic hypersensitivity pneumonitis (CHP), sarcoidosis, and infection/airway disease; (2) diagnosed with an indeterminate UIP pattern; (3) where IPF could not be confirmed at the MDD meeting or by clinical and imaging findings; (4) in an acute exacerbation status according to clinical condition and CT images; (5) with lung cancer and the sum of tumor diameters greater than 5 cm; (6) involving previous pulmonary surgery. The cases were divided into two groups: 80.0% for the training cohort, and 20.0% for the validation cohort. Patient characteristics, including age, gender, and smoking status, and pulmonary function (PFT; FVC%, FEV1%, DLCO%, and TLC%) were also recorded. The abbreviations are defined as: pulmonary function test (PFT), forced vital capacity (FVC%), forced expiratory volume (FEV1%), diffusing capacity of the lung for carbon monoxide (DLCO%), and total lung capacity (TLC%).

### 2.2. CT Imaging Acquisition Protocols

All CT studies were implemented with a 128-slice (SOMATOM Definition AS, Siemens Healthcare, Forchheim, Germany) or a 256-slice (SOMATOM Definition Flash, Siemens Healthcare) multi-detector CT (MDCT) scanner. All CT imaging protocols were identical and used the same acquisition parameters: scans with the collimation of 128 × 0.6 or 256 × 0.6 mm, tube voltage of 120 kVp, modulation of tube current, a gantry rotation speed of 0.5 s/r, and 1.5 mm reconstructed slice thickness in a single breath hold. Scan coverage was taken from the lung apex to the lowest hemi-diaphragm. All images were acquired in the supine position and at full inspiration status.

### 2.3. Image Interpretation

All CT images were reviewed by two radiologists (W.-M.H. and C.-H.Y.) with respectively 8 and 18 years of experience in chest CT and blinded to the clinical lung function information. The fibrotic score was obtained at six levels: (1) aortic arch, (2) 1.0 cm below the carina, (3) right pulmonary venous confluence, (4) halfway between the third and fifth section, (5) 1.0 cm above the right hemi-diaphragm, and (6) 2.0 cm below the right hemi-diaphragm (Figure 1) [18]. The proportion of content with at least one feature among honeycombing, traction bronchiectasis (TB), subpleural reticulation, or ground glass opacity with TB in each section, was scored to the nearest 5.0%, and the fibrotic score was measured as the average percentage among the above six sections [19].

### 2.4. State-of-the-Art Algorithm for Whole-Lung Parenchyma CT Analysis

All CT images were transferred to a dedicated artificial intelligence (AI) platform (QUIBIM Precision 2.8, QUIBIM SL, Valencia, Spain) and whole-lung CT segmentation was activated and performed automatically together with further 3D radiomics analysis and emphysema percentage evaluation, following DICOM (Digital Imaging and Communications in Medicine) images successfully received by the AI platform. The automatic activation of quantification analyses was based on the AI platform engine rule configured to verify if image DICOM tags matched pre-defined configurations for low-dose CT examination. The 2-class U-Net-based Convolutional Neural Network (CNN) algorithm with deep supervision layers was used by the AI-based deep learning algorithm, which provided the segmentation mask of right and left lungs and could be used in radiomics studies to establish correlations between textural features and clinical endpoints, such as diagnosis, prognosis, or treatment responses. This module is also equipped with a lung color map functionality for disease visualization, in which the lungs are segmented into Housefield Units (HU) and normalized on a [0,1] range, after which a jet colormap is applied to the normalized values. The deep learning algorithm not only performs lung mask extraction, but also provides quantitative data on radiomics features. The non-invasive, post-processing techniques are designed for quantifying features related to lung tissue heterogeneity. The extracted parameters can be classified as first-order textural features if they are obtained directly from the histogram, i.e., kurtosis and skewness, as well as second-order textural features if obtained from other techniques, such as correlation, entropy, contrast, and homogeneity.

### 2.5. Statistical Analyses

All statistical analyses were performed with IBM SPSS v22. The correlation between lung functions, fibrotic score, emphysema percentage, and radiomics features was tested using Spearman’s correlation. Binary logistic regression was also used for univariate and multivariate analyses. The results are presented as an odds ratio (OR) with a confidence interval (CI) of 95%. Lung function parameters are expressed as percentiles from normal predicted values. The inter-rater reliability of the fibrotic scores was assessed with intraclass correlation coefficients (ICC). All tests were two-sided, and *p*-values < 0.05 were considered significant.

The diagnostic accuracy of optimal predictive parameters was evaluated by the area under the curve (AUC) from receiver operating characteristic (ROC) analyses, and diagnostic sensitivity and specificity were calculated.

## 3. Results

We enrolled 556 cases for further evaluation in this study, of which 190 cases were excluded due to an alternative IPF diagnosis, and 52 cases were excluded due to the presence of with infection/inflammation or significant pleural effusion. Of the remaining cases, 103 were excluded because only subtle lung fibrosis was present and IPF could not be confidently diagnosed. Eight cases were excluded because of the presence of a lung cancer greater than 5.0 cm in size; 5 cases were excluded because of previous pulmonary surgery; 82 cases were excluded for the lack of standard chest CT images. Finally, 116 cases were retained for the investigation (90 non-cancer cases and 26 lung cancer cases); the cases were divided into a training cohort (79.3%, 92 cases) and a validation cohort (20.7%, 24 cases) (Figure 2).

### 3.1. Basic Characteristics

The basic characteristics of our population are shown in Table 1. Initial evaluation showed males accounted for 62.2% of the non-cancer cases and 92.3% of the cancer cases. There were 28.9% of the patients with a smoking status in the non-cancer group and 65.4% in the cancer group. The mean age of the two groups showed no significant difference (73.9 vs. 71.0 years). The mean lung volume was 3080.7 mL for the non-cancer group and 3952.6 mL for the cancer group; this difference was significant (*p* < 0.001). The fibrotic score was significant higher in the non-cancer group than that in the cancer group (21.7 vs. 16.3, *p* = 0.038). There was no significant difference in mean FVC (%), DLCO (%), FEV1 (%), TLC (%), or emphysema (%) between these two groups.

### 3.2. Inter-Rater Reliability of the Fibrotic Score

The fibrotic scores of chest CT upon initial examination were used for evaluating inter-rater reliability. Forty chest CTs were evaluated by two radiologists. The intraclass correlation coefficient (ICC) was 0.91 (*p* < 0.001), with an ICC > 0.9 considered as excellent.

### 3.3. Radiomics Feature Selection for Classifying Cancer and Non-Cancer Groups

Among 26 (Table 2) first-order and second-order key radiomics features of the Grey Level Co-occurrence Matrix (GLCM), energy (2.11 × 10^12^ vs. 2.73 × 10^12^) and kurtosis (18.81 vs. 22.99) showed significant differences between non-cancer and cancer patients in the training cohort (*p* < 0.001, and *p* = 0.029, Figure 3). Generally, LC had higher tendency to be associated with male and smoker patients, and HRCT revealed a greater tendency of developing LC with increasing energy and kurtosis (Figure 4). In the univariate logistic regression analysis model, LC development showed a significant association with the features of smoking, energy, and kurtosis, while in the multivariate logistic regression analysis model, there was a significant difference for energy (OR = 1.02, *p* = 0.03) and smoke (OR = 3.22, *p* = 0.04), but there was no significant difference for kurtosis (Table 3).

### 3.4. Diagnostic Accuracy

Diagnostic accuracy was evaluated by the AUC from ROC curves (Table 4). In the training cohort, the selected radiomics features (energy and kurtosis) showed an AUC of 0.66–0.73, sensitivity of 80.0–85.0%, and specificity of 54.2–59.7% (Figure 5A). Traditional risk factors, such as gender, smoking status, and emphysema (%), showed an AUC of 0.66–0.67, sensitivity of 55.0–90.0%, and specificity of 41.7–77.8% (Figure 5B). The combination of radiomics features and traditional risk factors produced an AUC of 0.79 and accuracy of 81.5%, better than the values found when using traditional risk factors only (AUC: 0.74 and accuracy: 77.2%). In the validation cohort, the selected radiomics features (energy and kurtosis) showed sensitivity of 83.0%, specificity of 38.9–44.4%, and accuracy of 50.0–54.2%. The traditional risk factors showed sensitivity of 33.3100%, specificity of 27.8–61.1%, and accuracy of 45.8–58.3% (Table 5). The combination of radiomics features and traditional risk factors produced diagnostic accuracy of 0.87 AUC and accuracy of 75.0%.

## 4. Discussion

Quantitative assessment has been increasingly used in the evaluation of IPF, especially in relation to PFT and disease progression [20,21,22,23]. Computer-aided detection of percentage fibrosis extent also help the prediction of disease-free survival for patients with lung cancer [24]. However, no previous reports used whole-lung texture analysis for risk stratification of IPF patients for LC. Radiomics, due to its high objectivity, has a potential as a supportive imaging-based tool, offering more detailed and reliable quantitative lesion assessment.

In this study, selected radiomics features—energy and kurtosis—could predict cancer-associated IPF with sensitivity of 80.0%–85.0% and specificity of 54.2–59.7%. The combination of radiomics features and traditional risk factors produced an AUC of 0.79 and accuracy of 81.5%, better than those for the traditional risk factors only (AUC: 0.74 and accuracy: 77.2%). In the validation cohort, the combination of radiomics features and traditional risk factors produced diagnostic accuracy of 0.87 AUC and accuracy of 75.0%. It was clinically proved that this 3D whole-lung texture analyses can achieve a more precise risk stratification of cancer-associated IPF.

According to radiomics feature analyses, cancer-developing IPF patients showed significantly greater energy and kurtosis (*p* < 0.001 and *p* = 0.029). Energy is the measure of the magnitude of voxel values in the image, and kurtosis relates to the peakedness of the distribution values. When evaluating tumor characteristics, kurtosis is related to intratumoral cellularity [25,26,27], and a high kurtosis value for a tumor indicates the homogeneity and denseness of the intranodular structure [28]. On the other hand, accumulated evidence shows that IPF and LC share common pathogenetic features [29]. IPF lung fibroblasts share many behaviors with cancer cells, including increased proliferation rates and resistance to apoptosis [30]. We speculate that lung fibroblasts in cancer-developing IPF patients are more likely to act as cancer cells than fibroblasts in IPF patients without cancer. Further, increased cellularity and density lead to higher kurtosis and energy. Of these two radiomics features, energy appeared more crucial and representative than kurtosis, as shown in both univariate and multivariate analyses, and showed better diagnostic accuracy in the training cohort.

We found that lung volumes were significantly lower in the non-cancer group who had a significantly higher fibrotic score. This could be due to the fact most patients in the cancer group were initially diagnosed as having lung cancer, and lung fibrosis was not as severe at the time. In contrast, patients in the non-cancer group typically showed no symptoms until fibrosis progressed to moderate or severe stages, leading to a decline of lung volume and worsening pulmonary fibrosis.

To our knowledge, this is the first study using whole-lung texture analyses to classify cancer-developing IPF. Because of the highly heterogeneous imaging patterns in this disease, whole-lung CT segmentation and analyses did not have the limitation of ROI selection and inter-observer variability. In addition, automatic lung parenchyma segmentation and analyses greatly saved manpower and time. This solution could also be used to determine disease severity and treatment response in follow-up studies.

This study has some limitations. First, the study population was relatively small, but with the covered over than 10 years. Further studies with a larger population are necessary. Second, we investigated cases of the past 11 years, and some of them did not undergo MDD for establishing a diagnosis. Although we excluded cases with ambiguous diagnosis, some cases may have been misdiagnosed. Third, we evaluated HRCT of patients with IPF combined with lung cancer, rather than before cancer development, due to the lack of such cases in our database. Although we limited the sum of tumor diameters to less than 5 cm to decrease the effect of tumor volume on whole-lung texture analysis, a basic difference between groups existed. Using image analysis before cancer development in future studies would be ideal.

## 5. Conclusions

Based on our results, whole-lung texture analysis provides a promising indicator for LC risk stratification of IPF patients. The combination of selected radiomics features and traditional risk factors, such as gender, smoking status, and emphysema percentage, can generate more accurate forecasts and provide more scientific evidence for diagnosing cancer-developing IPF.

## Figures and Tables

**Figure 1 cancers-13-05600-f001:**
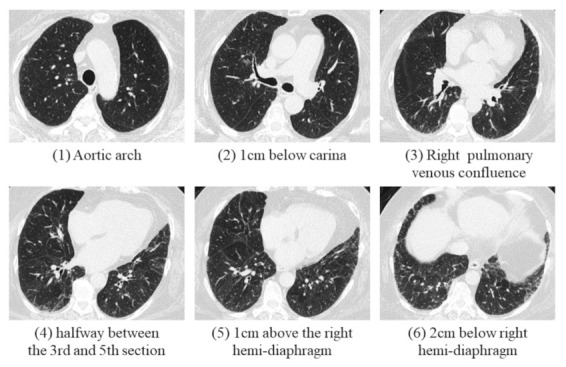
A 69-year-old female was diagnosed with IPF with probable UIP pattern in an MDD meeting. The percentage of fibrosis was calculated in these six levels, and the fibrotic score was calculated as the average percentage of these six sections.

**Figure 2 cancers-13-05600-f002:**
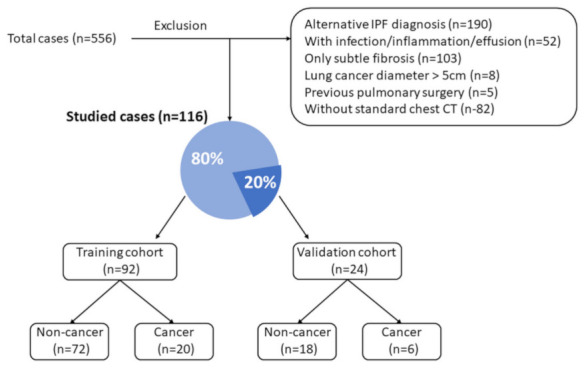
Patient selection flowchart for the training and validation cohorts.

**Figure 3 cancers-13-05600-f003:**
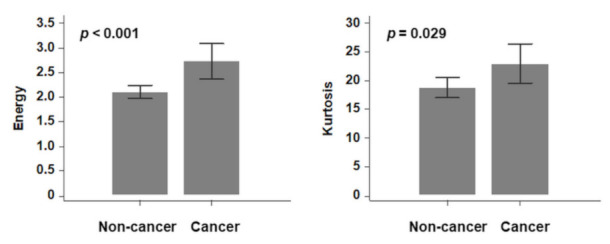
Radiomics features (energy and kurtosis) with significant differences between cancer and non-cancer groups.

**Figure 4 cancers-13-05600-f004:**
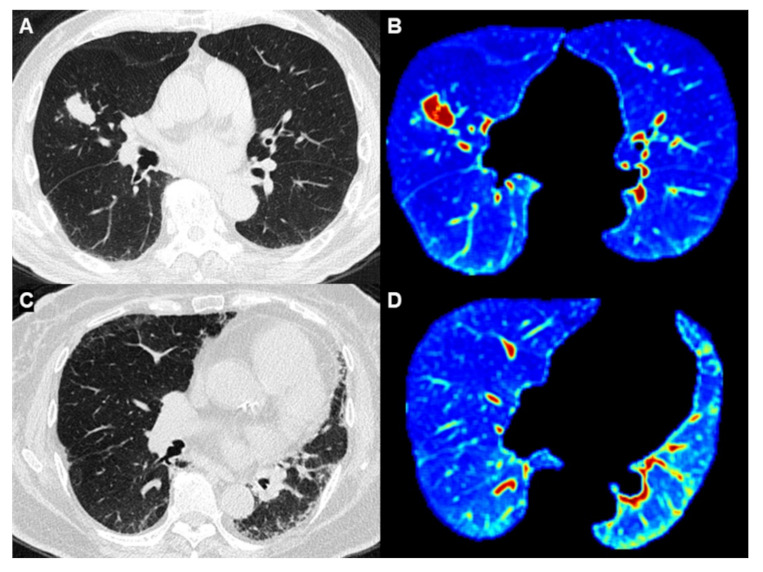
Comparison between IPF patients with and without lung cancer. (**A**,**B**) Images from an 87-year-old man who was a heavy smoker with IPF. (4A) HRCT showed an adenocarcinoma in the middle lobe of the right lung. (**B**) Whole-lung texture analysis revealed energy: 2.56 × 10^12^ and Kurtosis: 29.13 (**C**,**D**) Image from a 74-year-old woman who was a nonsmoker and was diagnosed with IPF. (**C**) HRCT showed pulmonary fibrosis without lung cancer (**D**) Whole-lung texture analysis revealed energy: 1.21 × 10^12^ and Kurtosis: 17.62.

**Figure 5 cancers-13-05600-f005:**
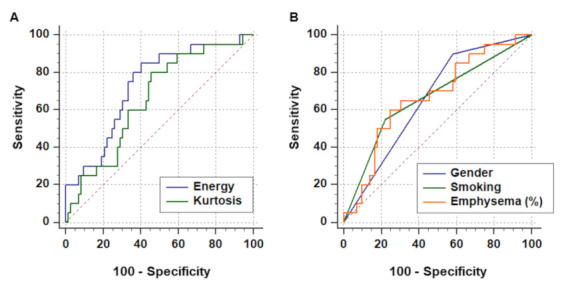
ROC curve for differentiating cancer-associated and non-cancer IPF. (**A**) Radiomics features (energy and kurtosis) demonstrated acceptable performance, with an AUC of 0.66–0.73, which was not inferior to (**B**) the performance of traditional risk factors (gender, smoke, and emphysema), with an AUC of 0.66–0.67.

**Table 1 cancers-13-05600-t001:** Basic characteristics.

Characteristics	Non-Cancer Group (*n* = 90)		Cancer Group (*n* = 26)		*p* Value
Age	73.9 ±	8.6	71.0 ±	10.4	0.153
Gender (M)	56	(62.2%)	24	(92.3%)	<0.001 *
Smoke	26	(28.9%)	17	(65.4%)	0.001 *
^1^ FVC (%)	87.0 ±	24.6	94.8 ±	22.1	0.239
^2^ FEV1 (%)	90.2 ±	26.6	92.6 ±	21.5	0.732
^3^ DLCO (%)	64.4 ±	22.6	56.6 ±	28.5	0.373
^4^ TLC (%)	78.7 ±	19.1	81.1 ±	12.1	0.628
Fibrotic score	21.7 ±	11.5	16.3 ±	11.5	0.038 *
Emphysema (%)	6.7 ±	6.1	9.00 ±	7.00	0.101
Lung volume (mL)	3080.7 ±	943.5	3952.6 ±	930.4	<0.001 *

^1^ FVC: Forced Vital Capacity; ^2^ FEV1: Forced Expiratory Volume; ^3^ DLCO: Diffusing Capacity of the Lung for. Carbon Monoxide; ^4^ TLC: Total Lung Capacity. ***** with significant difference.

**Table 2 cancers-13-05600-t002:** Comparisons of and representative CT radiomics features of non-cancer and cancer groups in the training cohort.

Metric	Features	Non-Cancer Group	Cancer Group	*p* Value
First order	Energy	2.11 × 10^12^	2.73 × 10^12^	<0.001 *
	Entropy	8.83	8.66	0.283
	Kurtosis	18.81	22.99	0.029 *
	Skewness	4.54	5.11	0.054
	Mean	−392.59	−415.87	0.574
	Standard deviation	401.30	377.47	0.092
	Median	−517.07	−521.57	0.905
	10th percentile	−716.30	−717.56	0.97
	90th percentile	105.02	−2.07	0.135
	Autocorrelation	650.74	614.40	0.507
Second order (^1^ GLCM)	Cluster Prominence	754,964.35	728,921.73	0.696
	Cluster shade	12,719.60	11,644.99	0.268
	Contrast	84.35	78.68	0.381
	Correlation	1.27	1.22	0.115
	Difference Entropy	7.12	7.05	0.556
	Difference Variance	46.55	43.58	0.347
	Dissimilarity	8.43	8.18	0.502
	Homogeneity	0.70	0.70	0.992
	^2^ IMC1	−0.15	−0.13	0.054
	^2^ IMC2	1.29	1.22	0.064
	Inverse difference	0.70	0.70	0.992
	Maximum probability	0.04	0.04	0.667
	Sum average	66.76	65.56	0.665
	Sum entropy	10.32	10.11	0.221
	Sum of squares	119.17	103.23	0.086
	Sum variance	392.33	334.26	0.073

^1^ GLCM: Grey level co-occurrence matrix. ^2^ IMC: Information Measure of Correlation. * with significant difference.

**Table 3 cancers-13-05600-t003:** Univariate and multivariate logistic regression model to differentiate cancer-developing ILD from non-cancer ILD in the training cohort.

Characteristic	Univariate Regression Analysis			Multivariate Regression Analysis		
	OR	(95% CI)	*p* Value	OR	(95% CI)	*p* Value
Smoke	4.28	(1.51–12.12)	0.006	3.22	(1.05–9.87)	0.041 *
Energy	1.52	(1.14–2.05)	0.001	1.02	(0.93–1.11)	0.012 *
Kurtosis	1.08	(1.01–1.15)	0.034	1.03	(0.95–1.11)	0.508

OR: odd ratio; CI: confidence interval. * with significant difference.

**Table 4 cancers-13-05600-t004:** Diagnostic accuracy based on gender, smoke, emphysema (%), and radiomics features in the training cohort.

Characteristics	Cut-Off	AUC	Sensitivity (%)	Specificity (%)	Accuracy (%)
Gender (M)		0.66 [0.55–0.75]	90.0	41.7	52.2
Smoke		0.66 [0.56–0.76]	55.0	77.8	72.8
Emphysema (%)	7.6	0.67 [0.56–0.76]	60.0	75.0	70.7
Energy	2.2 × 10^12^	0.73 [0.63–0.82]	85.0	59.7	69.6
Kurtosis	18.3	0.66 [0.55–0.75]	80.0	54.2	58.7

**Table 5 cancers-13-05600-t005:** Diagnostic accuracy in the validation cohort using cut-off values from the training cohort.

Characteristics	Sensitivity (%)	Specificity (%)	Accuracy (%)
Gender (M)	100	27.8	45.8
Smoke	100	44.4	58.3
Emphysema (%)	33.3	61.1	54.2
Energy	83.3	38.9	50.0
Kurtosis	83.3	44.4	54.2

## Data Availability

Data sharing not applicable.

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
