# Peer review of "Quantification of Cancer-Developing Idiopathic Pulmonary Fibrosis Using Whole-Lung Texture Analysis of HRCT Images"

_cancers, 2021, doi:10.3390/cancers13225600_

Round 1

Reviewer 1 Report

In this manuscript Liang and co-authors retrospectively evaluated radiomics features from HRCT in risk stratification for lung cancer in IPF patients.

In Table 1, is reported a significant difference in the fibrotic score between non-cancer group and cancer group (p=0.038) although the rate if the fibrotic score is equal between the two groups. This should be explained because it is attended that there is no difference between the two groups.

At line 91, the world “RT.” Should be removed.

In Table 2, and at line 170, the abbreviation (GLCM) should be specified.

Reviewer 2 Report

Recently I was requested to review a paper entitled Quantification of Cancer Developing Idiopathic Pulmonary Fibrosis Using Whole Lung Texture Analysis from HRCT. The paper is well written and easy to understand. The topic discussed in the manuscript is original and updated. I think it would be important to analyze innovative radiomic features in IPF patients to predict the increased risk of the development of lung cancer. However, I have some major and minor remarks that should be addressed prior to eventual publication.

Major remarks.

Line 79. The first inclusion criterion is confirmation of IPF by MDD. This was available during a short period of the study. It is not worth mentioning this factor as a first inclusion criterion. To be honest it cannot be an inclusion criterion at all. Please specify inclusion criteria. Please specify what were the methods of IPF diagnosis – how many were diagnosed on the basis of pathological examination. How were the patients diagnosed before MDD was introduced? It is essential information and must be included in the text. Going further as most of the patients were diagnosed with IPF before MDD did you consider critical reassessing of the diagnosis in order to exclude patients who would not fulfill the criteria for IPF diagnosis?

Patients diagnosed with lung cancer are more commonly smokers (Table 1.). Patients with lung cancer are also characterized by different radiomic features – kurtosis and energy (Figure 3). Please analyze whether a conclusion was drawn on the basis of a wrong premise. Please consider performing multivariate analysis to avoid any misinterpretation. Perhaps other radiographic features are not typical for lung cancer patients, but for smoking patients.

Minor remarks.

Line 157. It is sufficient to enter the value to one decimal place. Please consider making uniform changes to the text.

Line 160. Define what value is presented in the brackets.

Table 1. Define what value is presented in the brackets.

Round 2

Reviewer 2 Report

I would like to thank the authors for their efforts. I accept the paper for publication.